# Micro-Motion Parameter Extraction of Multi-Scattering-Point Target Based on Vortex Electromagnetic Wave Radar

**Lijun Bu †, Yongzhong Zhu \*,†, Yijun Chen †, Xiaoou Song †, Yufei Yang † and Yadan Zang †**

College of Information Engineering, Engineering University of PAP, Xi'an 710086, China
* Correspondence: haiyihaiyi@hotmail.com
† These authors contributed equally to this work.

**Abstract:** In addition to traditional linear Doppler shift, the angular Doppler shift in vortex electromagnetic wave (VEMW) radar systems carrying orbital angular momentum (OAM) can provide more accurate target identification micro-motion parameters, especially the detailed features perpendicular to the radar line-of-sight (LOS) direction. In this paper, a micro-motion feature extraction method for a spinning target with multiple scattering points based on VEMW radar is proposed. First, a multi-scattering-point spinning target detection model using vortex radar is established, and the mathematical mechanism of echo signal flash shift in time-frequency (TF) domain is deduced. Then, linear Doppler shift is eliminated by interference processing with opposite dual-mode VEMW. Subsequently, the shift in TF flicker is focused on the reference zero frequency by the iterative phase compensation method, and the number of scattering points is estimated according to the focusing effect. After this, through the constructed compensation phase, the angular Doppler shift is separated, then the angular velocity, rotation radius and initial phase of the target are estimated. Theoretical and simulation results verify the effectiveness of the proposed method, and more accurate rotation parameters can be obtained in the case of multiple scattering points using the VEMW radar system.

**Keywords:** vortex electromagnetic wave; micro-motion parameters; angular Doppler effect; spinning target; compensation phase

## 1. Introduction

Over the past few years, vortex electromagnetic waves have attracted increasing attention in the fields of rotation Doppler parameter estimation [1,2], OAM modulated wireless communication [3,4], and quantum applications [5] due to their unique spiral phase structure and orthogonal characteristics [6]. In addition, when a moving target is irradiated by vortex radar, a unique wave front shows the characteristic of angular diversity and the vortex echo contains more target information, which allows electromagnetic vortex radar to be utilized to improve the accuracy of object identification [7,8].

Traditional plane electromagnetic wave Doppler radars [9,10] use the linear Doppler effect generated by the relative motion between the radar and target to detect the micro-motion parameters of the target. However, radar detection based on conventional planar or spherical electromagnetic waves is ineffective when the target has a rotational motion perpendicular to the LOS direction. Due to the OAM-carrying property of the VEMW, the special helical phase structure can simultaneously detect the linear and angular Doppler effect (called the rotational Doppler effect), which provides a new method for extracting the micro-motion features of rotating objects [11].

In 1997, Courtial et al. used millimeter waves carrying OAM to hit rotating targets and observe the angular Doppler effect for the first time [12]. Subsequently, Lavery et al. successfully measured the relationship between the rotation frequency and Doppler shift by analyzing the OAM of light scattered from a spinning object [13]. Benefiting from the discovery of the vortex beam, a large number of studies on rotational Doppler have been

carried out successively. In [14], the authors proposed an indirect method based on phase measurement to detect the relationship between the Doppler frequency shift and angular velocity, which provides theoretical guidance for various application technologies of target recognition based on VEWM. Reference [15] takes a three-blade fan as the target to be tested, then uses a ring traveling wave slot antenna to generate an OAM beam for irradiation. By comparing the echo signals in different modes, a fixed value of the angular Doppler shift $l\Omega/2\pi$ can be verified. In [16], the authors proposed a method that can detect an object rotating around the beam propagation axis, which can both accurately estimate the rotation velocity and obtain the rotation direction of the target. A detection model of rotating targets under strabismus vision is established, and the relationship between the Doppler frequency shift, tilt angle, and rotation speed is deduced [17]. The Doppler effect and micro-Doppler effect of vortex electromagnetic waves were studied in [18], with the extraction method of angular Doppler frequency shift and micro-motion parameters in special scenes being deduced. However, due to the coupling between linear Doppler and angular Doppler, the angular Doppler needs to be separated. In [19], a method is proposed to detect the linear and angular acceleration of the object simultaneously; the echo signal is processed by time-frequency analysis, and the acceleration is decoupled through multiple OAM modes, laying the foundation for finding the three-dimensional velocity of the subsequent complex target detection. Despite all of these studies, the angular Doppler extraction method and parameter extraction of targets composed of multiple scattering points have not been studied. In [20], a signal separation method based on parametric demodulation was used to separate the angular Doppler of a cone target under forward-looking conditions; then, the Doppler information was used to estimate the micro-motion parameters and geometric feature. However, the algorithm is limited to a specific cone target scene, and the Doppler information of the separated targets is not in the same motion plane.

In conclusion, most of the existing parameter extraction algorithms based on VEMW have been carried out based on the premise of a single scattering point or a known number of scattering points, which limits the application of the angular Doppler effect in actual scenes. In this context, the present paper proposes a method for angular Doppler separation and rotating target parameter estimation based on vortex electromagnetic wave radar. First, according to the rotating target detection model of multiple scattering points, the instantaneous angular Doppler frequency shift is deduced. Then, the line Doppler and angular Doppler information of the target are separated by dual-mode echo interference processing. Time-frequency (TF) analysis is implemented on the angular Doppler information, and the slow time distribution of the Doppler frequency is obtained. Subsequently, according to the TF domain flash characteristics caused by the periodic rotation of the target, the phase compensation method is utilized to complete the separation of the angular Doppler information. Finally, the parameters of the rotating target are estimated by combining the separated angular Doppler and flash distribution. Subsequent simulation results verify the effectiveness of the algorithm. Compared with the existing algorithm, the advantage of this algorithm is that when the vortex echo contains multiple scattering point information, the Doppler information can be separated and used to estimate the number of scattering points, the rotation radius, the rotation angular velocity, and the initial phase.

The remainder of this paper is organized as follows: Section 2 introduces the scene and echo signal model of a spinning target; Section 3 describes the linear and angle Doppler separation, and derives the extraction method for the micro-motion parameters of multi-scattering-point target; the simulations and results are analyzed in Section 4; the discussion and conclusion are given in Sections 5 and 6.

## 2. Echo Model of Multi-Scattering-Point Target in VEMW Radar

The scenario geometry of spinning target detection is shown in Figure 1; in the figure, $O-XYZ$ is the radar coordinate system, and the transmitting antenna is located at the coordinate origin $O(0,0,0)$. The receiving antenna is composed of N array elements, which are placed equidistant in the $XOY$ plane, and the azimuth of the nth array element is $\varphi_n = n\frac{2\pi}{N}$,

$n = 0, 1, 2, \ldots, N - 1$. Suppose a multi-scattering-point object spins around the radar line of sight (LOS) direction with angular velocity, where $O'(x'_O, y'_O, z'_O)$ is the equivalent phase center and the corresponding polar coordinates are $(lllR_r, \theta_r, \varphi_r)$. Then, the unit vector in the LOS direction is $n_{LOS} = (n_x, n_y, n_z)^T = (\sin\theta_r \cos\varphi_r, \sin\theta_r \sin\varphi_r, \cos\varphi_r)^T$, the velocity vector of the scattering point P along the LOS is $v = (v_x, v_y, v_z)^T = v \cdot n_{LOS}$, the arbitrary angular velocity is $\omega = (\omega_x, \omega_y, \omega_z)^T = \omega \cdot n_{LOS}$, and its polar coordinate in the target coordinate system is $(r_p, \theta_0, \varphi_0)^T$.

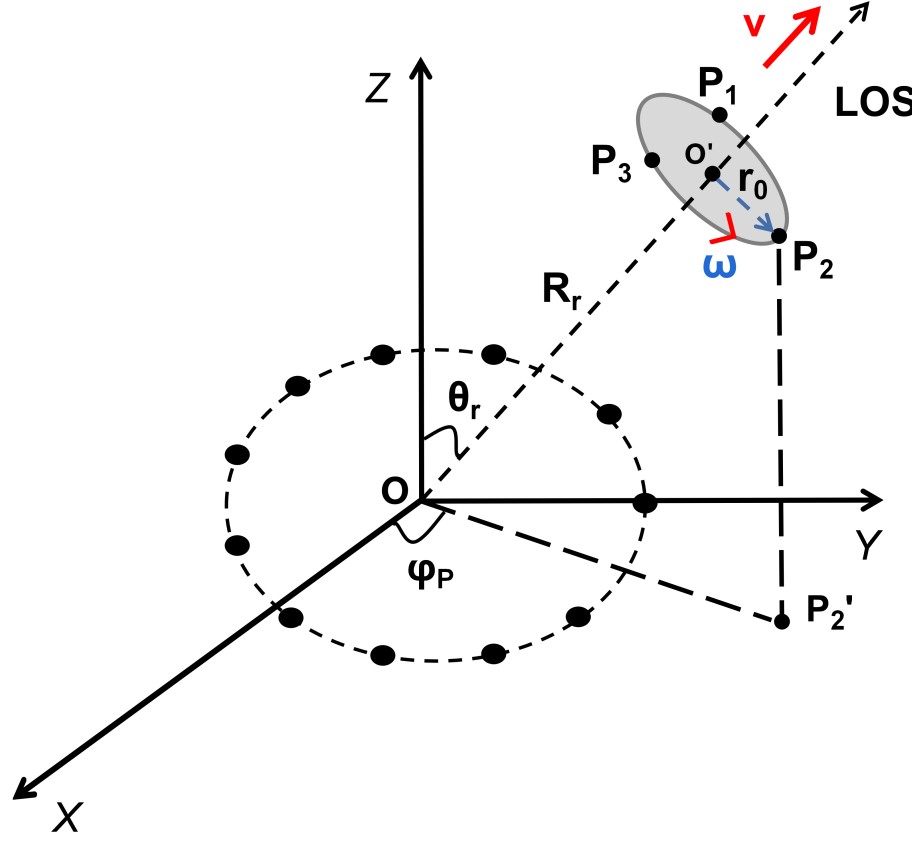

**Figure 1.** The detection model of a rotational object in VEMW-based radar system.

There are many ways to generate VEMW at present, such as nested double-arm helical antennas [21], helical phase plates [22], metasurfaces [23,24], etc. In this paper, the single-transmit multiple-receive method is used to generate VEMW carrying the mode of $l$ through the Uniform Circular Arrays (UCA). The received echo signal of the scattering point $P_2$ can be expressed as follows:

$$s(t, t_m, l) = \sum_{p=1}^{n} i^l \sigma_p N J_l \left[ k(t - \tau_p(t_m)) a \sin\theta_p(t_m) \right] \mathrm{rect}\left[ (t - \tau_p(t_m))/T_p \right] \exp\left[ il\varphi_p(t_m) \right]$$
$$\exp\left[ i2\pi \left( f_c(t - \tau_p(t_m)) + 0.5K(t - \tau_p(t_m))^2 \right) \right] \tag{1}$$

where $\sigma_p$ is the scattering coefficient of the scattering point, P; $f_c$ is the carrier frequency of the signal; $a$ is the radius of the UCA; $c$ is the speed of light; $J_l(\cdot)$ is the first kind of $l$-order Bessel function; $k(t) = 2\pi(f_c + Kt)/c$; $\tau_p(t_m) = 2r(t_m)/c$; and, among others, $r_p(t_m), \varphi_p(t_m), \theta_p(t_m)$ represent the distance from the scattering point to the antenna array, the azimuth angle, and the elevation angle, respectively. According to [2], the echo signal after dechirp can be expressed as:

$$
\begin{aligned}
s_{mix}(t,t_m,l) &= s(t,t_m,l) \cdot s_{ref}^{*}(t) \\
&= \sum_{p=1}^{n} i^l \sigma_p N J_l \left[k(t-\tau_p(t_m))a\sin\theta_p(t_m)\right] \mathrm{rect}\left[(t-\tau_p(t_m))/T_p\right]\exp\left[il\varphi_p(t_m)\right] \\
&\exp\left[-i2\pi\left(f_c\tau_p(t_m)-\tfrac{1}{2}K\tau_p^2(t_m)+K\tau_p(t_m)t\right)\right]
\end{aligned}
\tag{2}
$$

where the slow time $t_m$ is the pulse interval time sequence and the fast time $t$ is expressed as the pulse duration time sequence. In addition, it can be observed that the movement of the scatter point in the LOS direction generates the linear Doppler effect, which causes the phase term $\exp\left[-i2\pi\left(f_c\tau_p(t_m)-\tfrac{1}{2}K\tau_p^2(t_m)+K\tau_p(t_m)t\right)\right]$ to change. The movement in the direction perpendicular to the LOS generates the angular Doppler effect, which causes the phase term $\exp\left[il\varphi_p(t_m)\right]$ to change. Subsequently, the fast Fourier transform can be performed on the echo $s_{mix}(t,t_m,l)$ in the fast time $t$ dimension. It can be given as follows:

$$
\begin{aligned}
S(f,t_m,l) &= \sum_{p=1}^{n} i^l \sigma_p N \cdot H\left[f+K\tau_p(t_m)\right]\exp\left[il\varphi_p(t_m)\right] \\
&\exp\left[-i2\pi\left(f_c\tau_p(t_m)-\frac{1}{2}K\tau_p^2(t_m)\right)\right]\exp\left[-i2\pi(f+K\tau_p(t_m))\tau_p(t_m)\right]
\end{aligned}
\tag{3}
$$

where $H\left[f+K\tau_p(t_m)\right]=FFT[J_l(k(t))\,\mathrm{rect}(t/T_P)]$, the properties of which are similar to the $\sin c$ function [2]. The echo signal obtained after compensating for the residual video phase (RVP) term and the oblique term can be written as

$$
S_1(f,t_m,l)=\sum_{p=1}^{n} i^l \sigma_p N \cdot H\left[f+K\tau_p(t_m)\right]\exp\left[il\varphi_p(t_m)\right]\exp\left[-i2\pi f_c\tau_p(t_m)\right].
\tag{4}
$$

The phase term of the processed echo signal $S_1(f,t_m,l)$ can be expressed as $\phi(t_m)=-4\pi f_c r(t_m)/c+l\varphi_p(t_m)$, while the theoretical Doppler frequency shift of the echo signal can be derived as follows [17]:

$$
f_D=\frac{1}{2\pi}\frac{d\phi(t_m)}{dt_m}=-\frac{2f_c}{c}\frac{dr(t_m)}{dt_m}+\frac{l}{\pi}\frac{d\varphi(t_m)}{dt_m}=f_L+f_A
\tag{5}
$$

where $f_L$ is the line Doppler shift and $f_A$ represents the angular Doppler shift of the scattering point. In order to observe the change rule of Doppler shift, the coordinate vector of scatterer P is analyzed:

$$
\begin{aligned}
OP_p &= \left(x_p(t_m),y_p(t_m),z_p(t_m)\right)^T \\
&= \mathrm{Rinit}\left\{\begin{array}{c} r_p\cos(2\pi\Omega t_m+\psi_p) \\ r_p\sin(2\pi\Omega t_m+\psi_p) \\ 0 \end{array}\right\}+\left\{\begin{array}{c} v_x+x_{O'} \\ v_y+y_{O'} \\ v_z+z_{O'} \end{array}\right\}
\end{aligned}
\tag{6}
$$

where $r_p,\Omega,\psi_p$ represent the rotation radius, the rotation frequency, and the azimuth angle at initial moment, respectively, and Rinit is the Euler matrix. From the polar coordinates of the equivalent phase center $O'$, the initial Euler angles are $(0,-\theta_r,-\varphi_r)$. In order to facilitate the analysis of the relationship between Doppler frequency shift and the spin parameters, the approximate model of the angular Doppler frequency shift can be deduced as according to [25]

$$
f_A=\frac{l}{\pi}\cdot\frac{2\pi r_p\Omega\cos(2\pi\Omega t_m+\beta+\psi_p)}{(R_0+vt_m)\sin\theta_p}
\tag{7}
$$

where $\beta=\arctan\left[\tan\theta_0\left(n_y\cos\varphi_0-n_x\sin\varphi_0\right)\right]$. According to (4) and (7), the following conclusions can be obtained:

(1) The Doppler shift of the echo signal includes the linear Doppler frequency shift induced by the distance variation of multiple scattering points along the LOS and the angular Doppler frequency shift caused by the translation and rotation motion. For the angular Doppler shift, it is caused by the motion when the two parts are coupled with each other.

(2) The magnitude of the angular Doppler frequency shift is positively related to the transmitted OAM mode $l$, the rotation radius $r_p$, and the rotation frequency $\Omega$, which is irrelevant to the signal carrier frequency $f_c$ and the bandwidth B.

(3) When the translational component of the target is small or zero, the angular Doppler frequency shift caused by the rotational motion appears as a sinusoidal curve in the TF domain. The angular Doppler shift model at this time can be expressed as

$$f_A = A_p \cos(2\pi\Omega t_m + \beta + \psi_p) \tag{8}$$

Because the focus of this paper is on the micro-motion feature extraction of rotating targets, the translational velocity of the target in the LOS direction can be estimated using existing mature algorithms [26] or by transmitting plane electromagnetic waves as auxiliary signals to achieve the corresponding velocity estimation [27]. In the following analysis, it may be assumed that the translational velocity of the target is known and small.

## 3. Separation of Doppler Effect

The emphasis of this paper is to extract the micro-motion features of the target by angular Doppler frequency shift. Thus, it is necessary to successively separate the angular Doppler information of a single scattering point according to the echo signals processed above.

### 3.1. Separation of Line Doppler Effect

Above all, the dual-mode interference processing method is used to separate the linear Doppler effect from the angular Doppler effect. According to (4), the dual-mode echo can be written as follows [2]:

$$\left.\begin{array}{l} S_1(f, t_m, l) = \sum_{p=1}^{n} i^l \sigma_p N \cdot H\big[f + K\tau_p(t_m)\big] \exp\big[-i2\pi f_c \tau_p(t_m)\big] \exp\big[il\varphi_p(t_m)\big] \\ S_2(f, t_m, l) = \sum_{p=1}^{n} i^{-l} \sigma_p N \cdot H\big[f + K\tau_p(t_m)\big] \exp\big[-i2\pi f_c \tau_p(t_m)\big] \exp\big[-il\varphi_p(t_m)\big] \end{array}\right\} \tag{9}$$

It can be found that the above two equations are consistent except for the azimuth term. The two signals of (9) are conjugated and multiplied, and when there is only a single scattering point in the distance unit, the processed echo signal can be obtained as

$$\begin{aligned} S_3(f, t_m, l) &= S_1(f, t_m, l) \cdot \text{conj}[S_2(f, t_m, -l)] \\ &= \sum_{p=1}^{n} \sigma_p^2 N^2 \exp\big[i2l\varphi_p(t_m)\big]\big|H(f + K\tau_p(t_m))\big|^2 \end{aligned} \tag{10}$$

At this time, the linear Doppler information in the echo signal is completely suppressed and the angular Doppler information of the target is enhanced, which can be extracted relatively accurately. However, the processed echo contains signal components of multiple scattering points, and the Doppler frequencies are aliased in the TF domain. If the parameter extraction method based on the TF image is used directly, it is very difficult to estimate the parameters with high precision. In order to obtain accurate estimation of the target parameters, a phase compensation method based on the amplitude flash distribution in the TF domain is used to separate the angular Doppler signals of different scattering points. The specific method is as follows:

Combining the definition of Doppler shift in (5) and the angular Doppler shift model derived in (8), the azimuthal model of the scattering point can be written as

$$\varphi_p(t_m) = \int f_A dt_m = \int A_p \cos(\omega t_m + \beta + \psi_p) dt_m$$
$$= \frac{A_p}{\omega} \sin(\omega t_m + \beta + \psi_p) \tag{11}$$

Ignoring the influence of the amplitude, the echo of a single scattering point in (10) can be abbreviated as

$$S_4(f, t_m, l) = \exp[i\Lambda \sin(\omega t_m + \alpha)] \cdot |H(f + K\tau_p(t_m))|^2 \tag{12}$$

where $\Lambda = 2l\frac{A_p}{\omega}$, $\alpha = \psi_p + \beta$; then, the processed echo is the product of the $\sin c$-like function and the phase term, and the echo Doppler frequency is time-varying. Therefore, the Short Time Fourier Transform (STFT) can be utilized to analyze it.

Figure 2 depicts the slow time domain echo of the distance unit where the target is located. From the properties of the FFT, it can be known that

$$S_5(f, f_m, l) = FFT\left\{\exp\left[i\Lambda \sin(\omega t_m + \beta + \psi_p)\right]\right\} \otimes FFT\left\{\left|H(f + K\tau_p(t_m))\right|^2\right\} \tag{13}$$

where the superscript $\otimes$ represents a convolution computation, the $\sin c$ function and rect$[f]$ are Fourier transform pairs, and

$$FFT\{\exp(j\Lambda \sin(\omega t_m + \alpha))\} = \frac{\Lambda w}{2\pi} \cos(\omega t_m + \alpha). \tag{14}$$

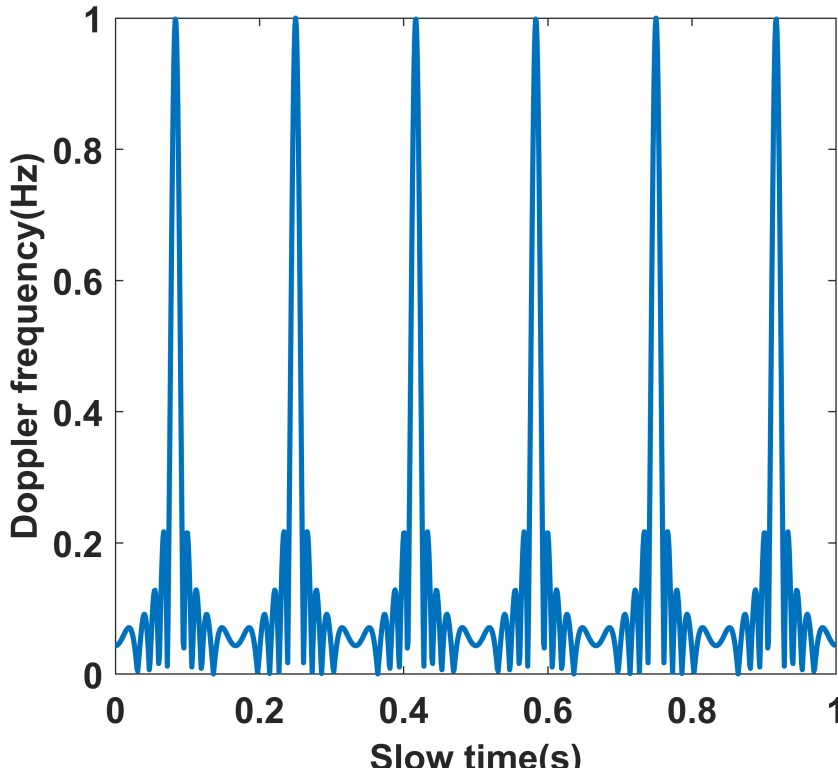

**Figure 2.** Echo signal in the slow time domain.

It can be seen that the Fourier transform of the echo envelope after processing is a stripe function with a certain bandwidth, which appears as a flash band in the TF domain, as shown in Figure 3a; on the other hand, the Fourier transform of the phase is in cosine

form, as shown in Figure 3b. The complete echo signal presents flash strips fluctuating with slow time in the TF domain, as shown in Figure 3c, and the color bars on the right side of the figures display color levels. Therefore, if the phase of a certain scatter point is compensated fully, it can be identified and separated from the multiple flashes generated by multiple scatter points. Furthermore, the flash focused on the reference zero frequency belongs to the same scatter point, and the number of scattering points can be estimated by the number of flashes between two zero-frequency blinking intervals [28].

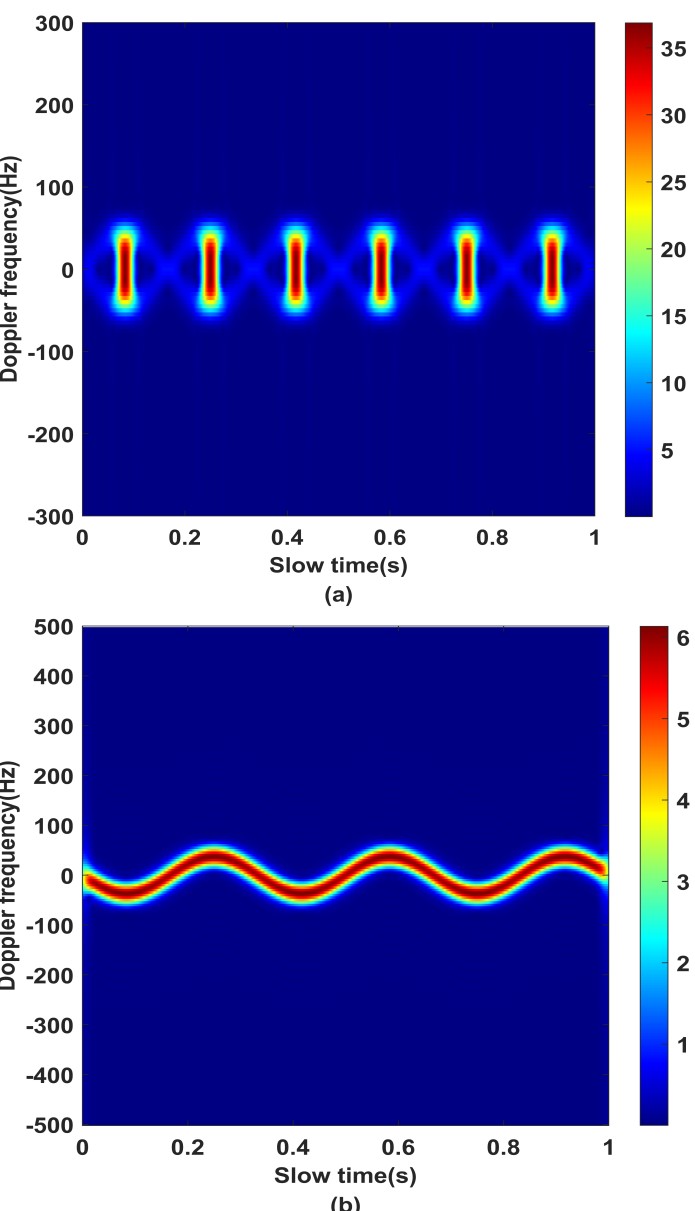

**Figure 3.** *Cont.*

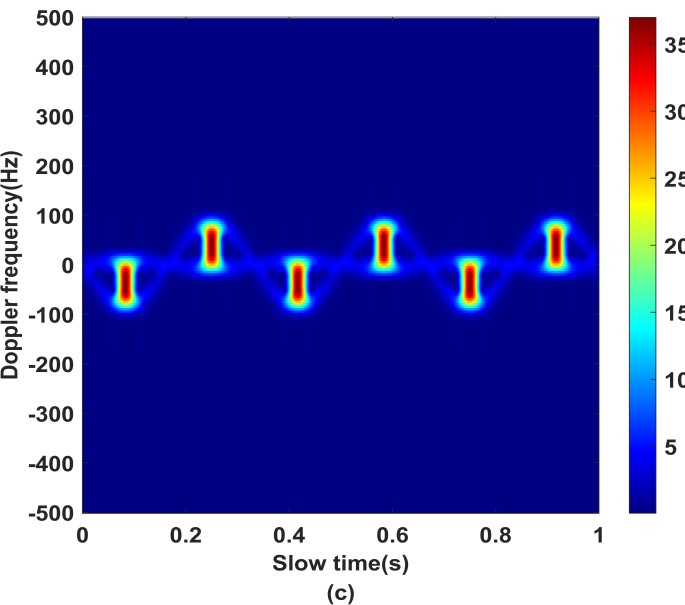

**Figure 3.** The echo of a single scatter in the time-frequency domain: (**a**) profile distribution, (**b**) phase distribution, (**c**) integrated echo signal.

### 3.2. Separation of Angular Doppler Effect

According to (12), the compensation phase model of the echo signal can be written as

$$S_{\text{com}} = \exp(-j\mu \sin(\eta t_m + \chi)) \tag{15}$$

where $\mu, \eta, \chi$ represent the amplitude, rotational angular velocity, and initial phase, respectively; as it can be known from (12) that $\Lambda = 2l\frac{A_p}{\omega}$, there is a definite relationship between $\mu$ and $\eta$. This transforms the separation of the angular Doppler effect into an optimization problem for the two-dimensional parameters $\eta, \chi$. Then, the compensated signal can be written as

$$\begin{aligned} S_6(f, t_m, l) &= S_4(f, t_m, l) \cdot S_{com} \\ &= \left| H(f + K\tau_p(t_m)) \right|^2 \cdot \exp[j\Lambda \sin(\omega t_m + \alpha) - j\mu \sin(\eta t_m + \chi)] \end{aligned} \tag{16}$$

The different phenomena of the phase term after compensation are analyzed below, including the following three situations:

(a) When $\mu = \omega, \chi = \alpha$, the azimuth term of the scattering point is fully compensated, as shown in Figure 4; then, (16) can be expressed as

$$\begin{aligned} S_6(f, t_m, l) &= S_4(f, t_m, l) \cdot S_{com} \\ &= \left| H(f + K\tau_p(t_m)) \right|^2 \end{aligned} \tag{17}$$

(b) When $\eta \neq \omega, \chi = \alpha$, the compensated phase is the difference between two sinusoidal signals with different frequencies, and it appears as an oscillation function of amplitude in the slow time, as shown in Figure 4a; then, (16) can be written as

$$\begin{aligned} S_6(f, t_m, l) &= S_4(f, t_m, l) \cdot S_{com} \\ &= \left| H(f + K\tau_p(t_m)) \right|^2 \cdot \exp[j\Lambda \sin(\omega t_m + \alpha) - j\mu \sin(\eta t_m + \alpha)] \end{aligned} \tag{18}$$

(c) When $\eta = \omega, \chi \neq \alpha$, the compensated phase is the difference between two sinusoidal signals with the same frequency and different initial phases, which is expressed as a cosine function in the slow time dimension, as shown in Figure 4b; then, (16) can be written as

$$S_6(f, t_m, l) = S_4(f, t_m, l) \cdot S_{com}$$
$$= \left| H(f + K\tau_p(t_m)) \right|^2 \cdot \exp\left[ j2\Lambda\left( \cos\left( \omega t_m + \frac{\alpha + \chi}{2} \right) \sin\left( \frac{\alpha - \chi}{2} \right) \right) \right]. \tag{19}$$

In summary, it can be seen that in the second case the compensated Doppler frequency shows an oscillating change with slow time; in the third case, the compensated Doppler frequency shows a cosine curve with slow time, and the corresponding flash cannot be focused on benchmark zero frequency in the TF domain. The target flash can be focused on the reference frequency only if the phase is fully compensated.

Through the above analysis, in order to avoid the problems of unsatisfactory focusing effect at the reference frequency and inaccurate parameter search estimation caused by Doppler frequency oscillation or phase mutation error, this paper considers parameter search estimation for all amplitude flashes within the imaging time and finds the optimal parameters from it. In addition, the processed echo contains the information of multiple scattering points. During the initial compensation, it is impossible to determine which amplitude flash belongs to the same scattering point.

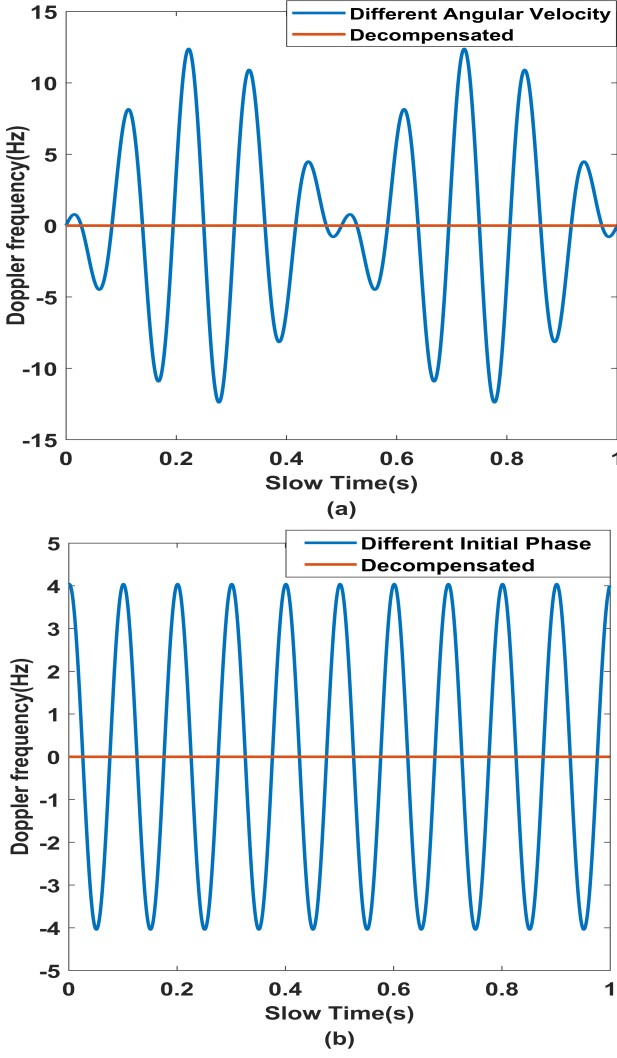

**Figure 4.** The result between the different sine functions after compensation: (**a**) compensated result of different angular velocity and (**b**) compensated result of different initial phase.

Here, the basic idea is to leverage the iterative estimation algorithm to select any flash, which is used as the observation object and the compensation object; then, the first iterative

compensation based on the principle of the minimum flash frequency is carried out, and after the corresponding TF diagram is obtained, the corresponding observation object in the second group of amplitude flash is found. All the amplitude flashes belonging to the same scattering point were searched in turn, and focused on the reference frequency, while the flashes of other scattering points deviated from the reference frequency. At this time, the number of scattering points were obtained according to the number of flashes between two adjacent observation objects, and the rotation angular velocity $\omega$ and initial phase $\psi_p$ of the target estimated according to the compensation phase constructed by the search. From (14), the maximum value of the Doppler frequency is $f_{\max} = \frac{A\omega}{2\pi}$; combined with $A_p = \frac{l}{\pi} \frac{r_p \omega}{R_0 \sin \theta_r}$ in (8) and (12), the radius of rotation can be written as

$$f_{\max} = \frac{l^2}{\pi^2} \frac{r_p \omega}{R_0 \sin \theta_r} \tag{20}$$

As can be seen from the above analysis, after obtaining the rotational velocity, the rotation radius $r_p$ of the scattering point can be estimated according to the Doppler frequency value of the flash in the TF domain. The flowchart of the proposed VEMW-based parameter estimation method is shown in Figure 5.

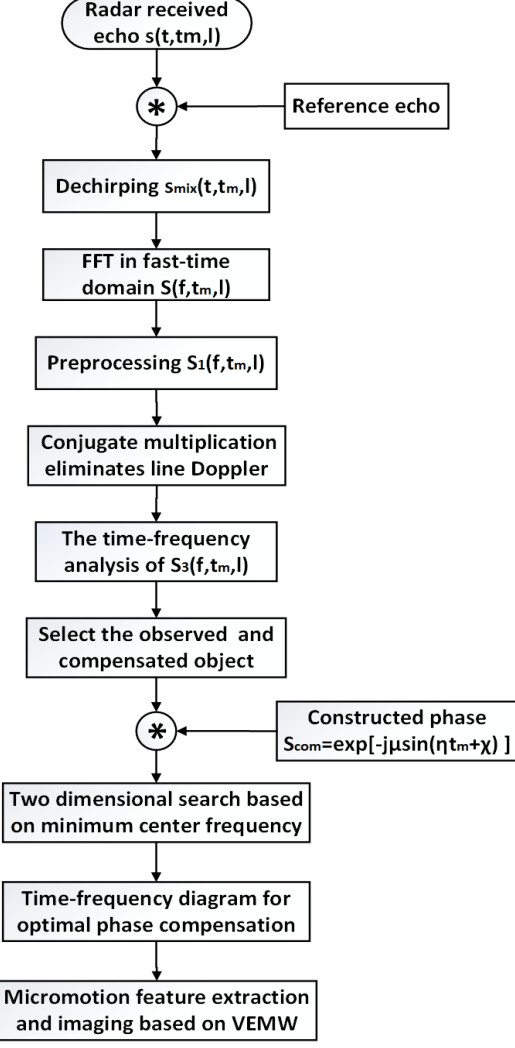

**Figure 5.** The flowchart of the proposed method.

## 4. Simulation Experiment and Result Analysis

In this section, a series of simulations and analysis of the results are presented to validate the effectiveness of the above-mentioned theoretical analyses.

### 4.1. Simulation Experiment

This section is used to prove the accuracy of the separation algorithm and the extraction method used to obtain the micro-motion parameters. The main parameters of the simulation are listed in Table 1.

**Table 1.** Key parameters in the simulation.

| Parameters | Values |
|---|---|
| Center frequency $f_c$ | 10 GHz |
| Pulse width $T_P$ | 80 μs |
| Bandwidth $B_r$ | 2 GHz |
| PRF | 5000 Hz |
| Target rotation velocity $\omega$ | $6\pi$ rad/s |
| OAM mode $l$ | 30 |
| Center of rotation $O'$ | (200 m, $6\pi/30$ rad, $\pi/3$ rad) |
| The radius of UCA | 0.5 m |
| Radius of rotation $r_p$ | 0.5 m |

The simulation results are shown in Figure 6, in which Figure 6a is the representation of the echoes in the distance unit with the scattering point in the slow time domain. Due to the scattering point rotating at a uniform speed, the echo signal appears as a series of spectral peaks with the same interval in the slow time domain. The STFT is performed on the echo sequence in the slow time domain in Figure 6a; the results are shown in Figure 6b as a series of positive and negative frequency flashes that vary with slow time alternately in the TF domain. The bandwidth of each flash is 83.92 Hz and the duration is 0.016 s.

In order to further determine the number of scattering points, the radius of rotation, and other parameters, the echo signal after linear and angular Doppler separation is processed in the slow time domain by the phase compensation method. Then, Figure 7 shows the TF domain image of the echo after iterative compensation. In the analysis process, we select the fourth flash in Figure 6b as the observation object, then choose the TF diagram with the best compensation effect according to the principle that the center frequency of the observation object is closest to the reference zero frequency.

Figure 7a shows the focusing effect obtained by the first phase compensation. It can be seen that the center frequency of the observed object is close to zero frequency compared with Figure 6b. In addition, other flashes have different degrees of frequency shift compared with Figure 6b. Figure 7b shows the focusing effect after the second iterative compensation. This phenomenon is caused by the difference between the angular frequency of the compensation phase constructed during the compensation process and the actual angular frequency of the rotating target. These results are consistent with the results of the amplitude oscillation of the Doppler frequency in the slow time domain after compensation in Figure 4b.

Figure 7c shows the TF spectra after the third iterations of compensation. In Figure 8, it can be seen that the center frequency of the observed object is already at zero frequency, and the other flash distributions are almost identical.

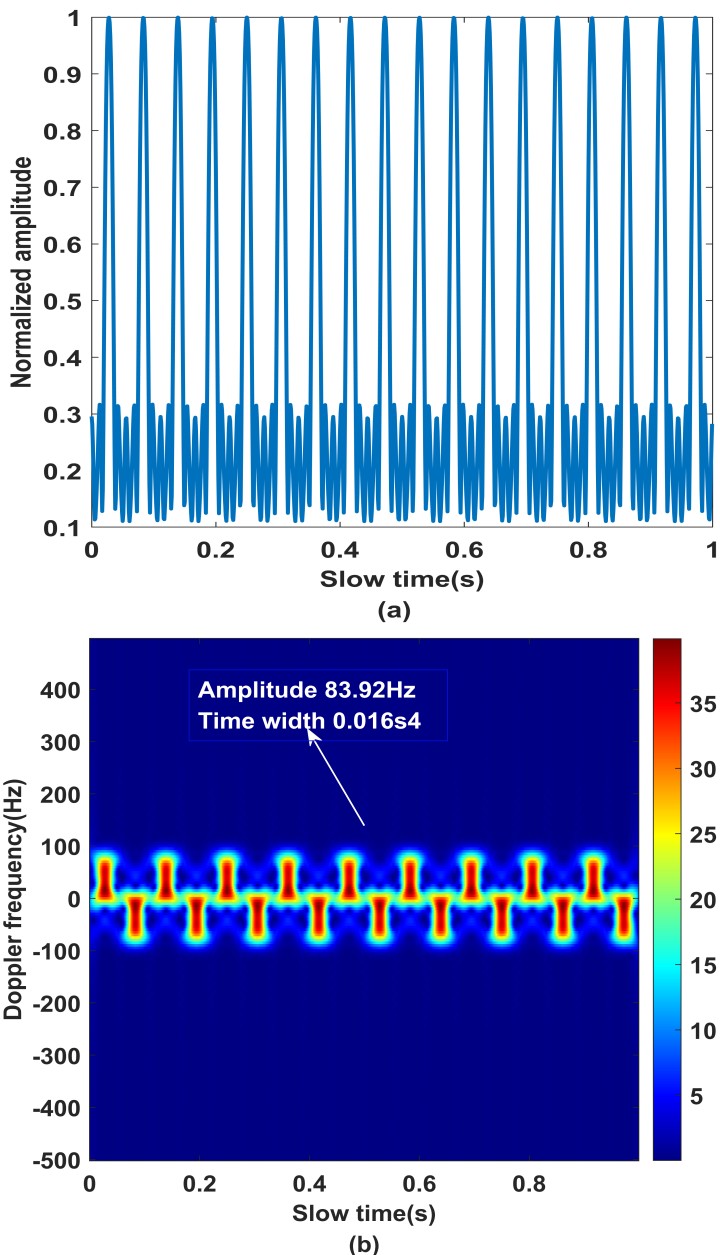

**Figure 6.** Echo signals of rotating target: (**a**) echo signal in the slow time domain and (**b**) echo signal in the time-frequency domain.

The flashes at the five different positions indicated by the arrows are all focused on the zero frequencies, which belong to the same scattering point of the rotating target and are generated in different slow times. At this time, the positions of the five flashes in slow time are as follows: 0.194 s, 0.361 s, 0.527 s, 0.694 s, 0.862 s. In addition, two flashes belonging to the same scatterer contain two flashes with different slow times and different Doppler frequencies. It can be judged that the rotating target contains three scattering points.

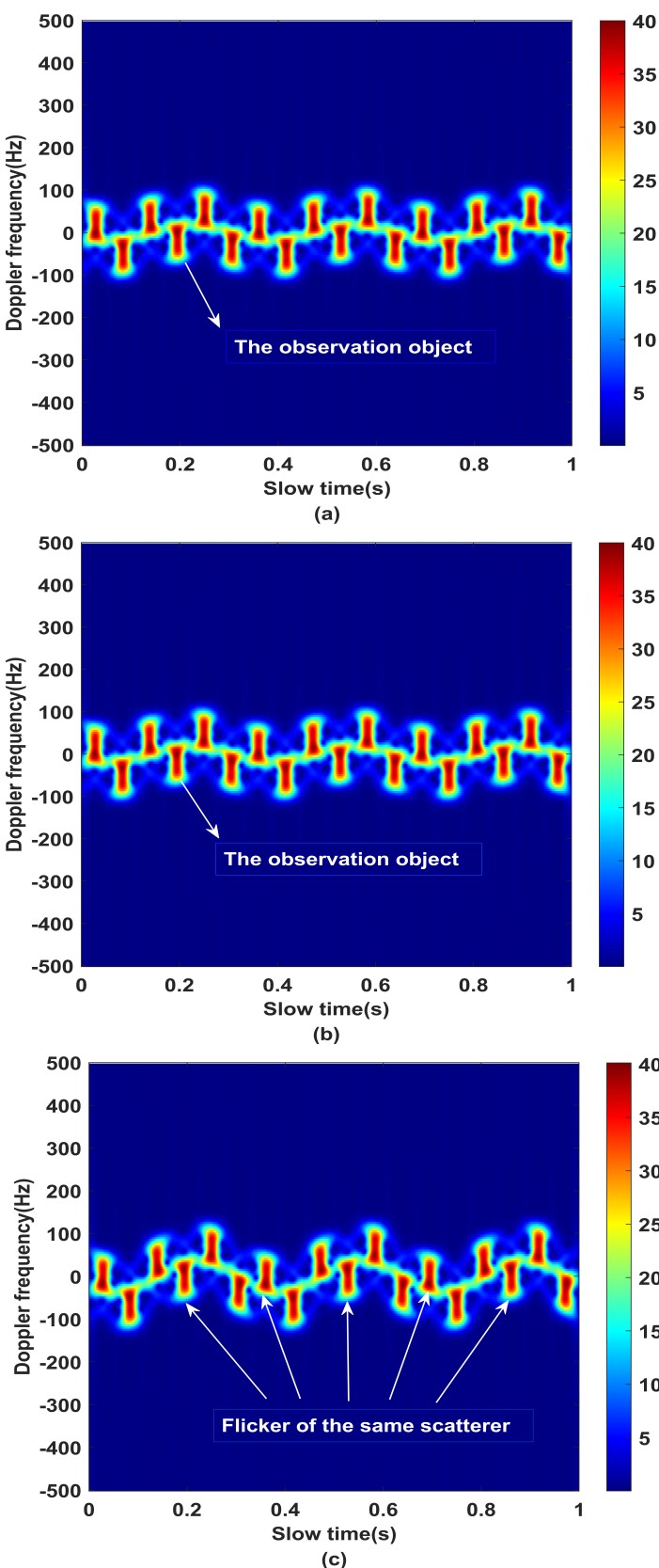

**Figure 7.** Time-frequency diagram of echo signal after iterative compensation: (**a**) obtained by first iteration, (**b**) obtained by second iteration, (**c**) obtained by third iteration.

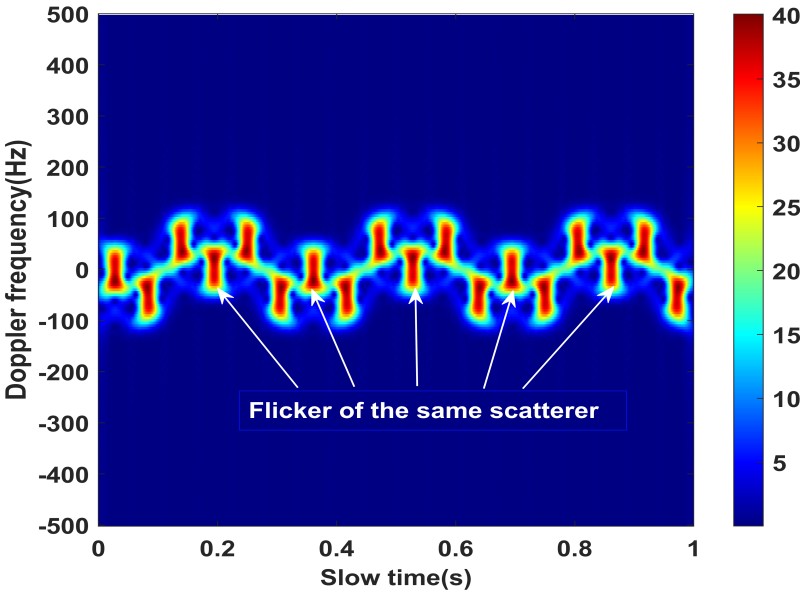

**Figure 8.** The best results obtained by the fourth iteration.

### 4.2. Effect Analysis of Angular Doppler Separation

The phase compensation algorithm proposed in this paper finds the flash of the scattering point focus on the reference zero frequency in the TF domain by completely compensating the echo phase. Then, the angular Doppler information is obtained according to the constructed compensation phase to realize the angular Doppler separation of the rotating target. By repeating the above compensation process, the information of each scattering point is obtained successively.

Taking the spinning target in the above simulation experiment as an example, after using phase compensation, the echo separation effect in the TF domain is shown in Figure 9. The flash with complete phase compensation is taken as the first scattering point, then the second scattering point (shown by the white curve arrow) and the third scattering point (indicated by the red arrow) are sequentially determined according to the time between each flash and the zero-frequency flash in the slow time dimension.

Figure 10 is the angular Doppler TF diagram of each scattering point obtained according to the constructed compensation phase. It can be seen that the angular Doppler frequency shift satisfies the sinusoidal variation rule.

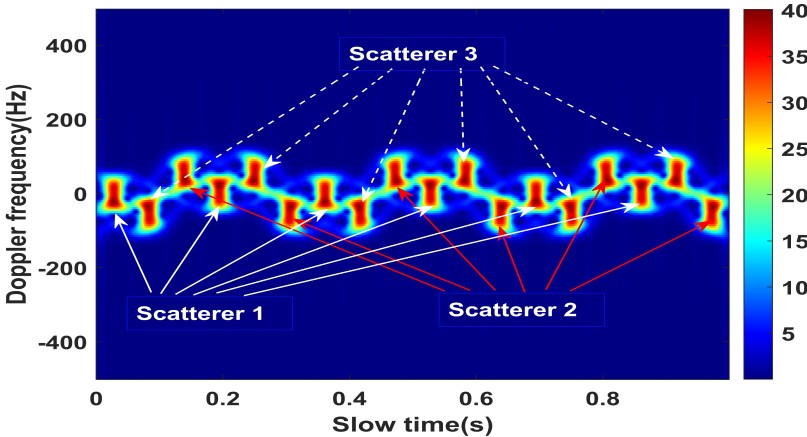

**Figure 9.** Scatterer separation in the time-frequency domain.

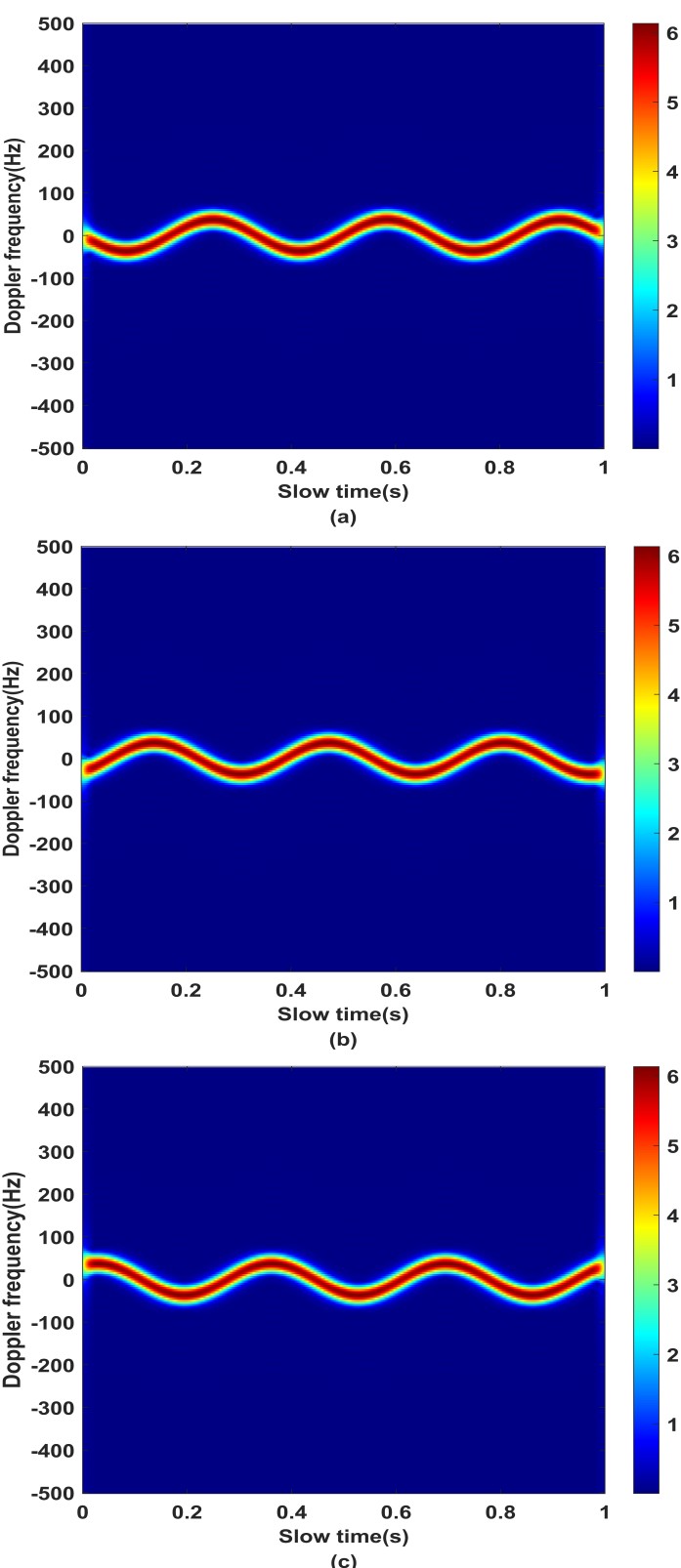

**Figure 10.** TF diagram of the angular Doppler: (**a**) angular Doppler of scatterer 1, (**b**) angular Doppler of scatterer 2, (**c**) angular Doppler of scatterer 3.

### 4.3. Parameter Estimation Accuracy Analysis

The above section has verified the effectiveness of the algorithm for angular Doppler information separation of rotating targets. In the following section, the parameter estimation

accuracy of scatterers is analyzed. As shown in Figure 6b, the compensated echo achieves the best focusing effect in the TF domain, and the construction phase corresponding to the fourth iteration compensation is

$$S_{com} = \exp(-j13.987\sin(6.12\pi t_m + 1.07)) \tag{21}$$

It can be seen that the characteristic parameters of the first scatter point are the rotational angular velocity, $\omega = 6.12\pi \mathrm{rad/s}$, and the initial phase of the scattering point, 1.07 rad. According to (20), the estimated value of the rotation radius can be obtained as $r_p = 0.5103$ m. According to Table 2, the average estimation error of the characteristic parameters is 2.14%. This shows that the parameter estimation method for a rotating target based on VEMW proposed in this paper can realize the effective estimation of characteristic parameters.

**Table 2.** Error analysis of parameter estimation results.

| Parameters | Real Value | Estimated Value | Evaluated Error |
|---|---|---|---|
| Scattering points $n$ | 3 | 3 | 0 |
| Angular velocity $\omega$ | $6\pi$ | $6.12\pi$ | 2.17% |
| Initial phase | 1.0472 rad | 1.07 rad | 2.18% |
| Radius of rotation $r_p$ | 0.5 m | 0.5103 m | 2.06% |

In addition, in order to further improve the accuracy of the proposed algorithm, the estimated parameters are analyzed below. First, the estimation of the number of scattering points is based on the positional relationship of each flash in the TF diagram, which determines whether the constructed phase function can fully compensate the echo signal. Next, the estimation of initial phase only affects the overall offset of the flashes, and does not determine the relative positions of each flash. In addition, the estimation of the rotation radius is mainly based on the rotation angular velocity in (20). Therefore, the estimation of angular velocity affects the compensation effect of the whole echo. Figure 11 shows the difference between the center frequency of the observed flash and the reference zero frequency in the TF diagram after phase compensation for the same initial phase with different angular velocities.

The average value was obtained after 100 Monte Carlo experiments; it can be seen that when the angular velocity is closer to $\omega = 6\pi \mathrm{rad/s}$, the flash frequency difference decreases more obviously, and the best compensation effect is obtained when the angular velocity of the compensation phase is equal to the actual angular velocity. Therefore, the algorithm based on phase compensation proposed in this paper can achieve more accurate angular Doppler separation and feature extraction of rotating targets.

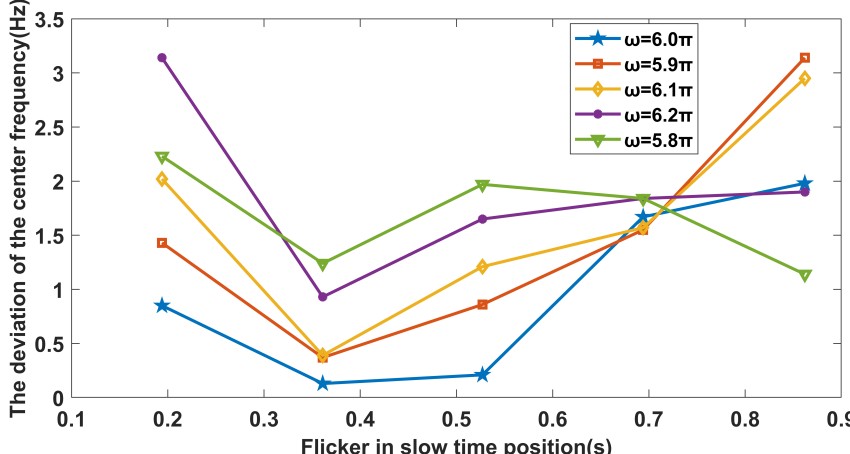

**Figure 11.** The effect of different angular frequency compensation phases on flicker position.

## 5. Discussion

The angular Doppler aliasing problem has not been considered in previous studies of parameter estimation methods for micro-moving targets using VEMW. It is noted in this paper that Doppler information is aliased in the process of feature extraction, as the rotating target contains multiple scatterers, a phenomenon that seriously affects the accuracy of parameter estimation. In order to overcome this problem, an angular Doppler separation and feature extraction method based on VEMW radar is proposed. In real applications, the velocity of the target in the LOS direction may be larger, which means that a more accurate angular Doppler shift model is needed for analysis. In this case, the existing Doppler model is not suitable. Future research may extend the echo signal model to consider both velocity and acceleration.

## 6. Conclusions

In general, in this paper we investigated the detection algorithm of the rotational target with multiple scatterers utilizing VEMW. By analyzing the characteristics of echo signals and flashes in the TF domain, an angular Doppler separation and characteristic parameter estimation method is proposed based on phase compensation. First, the linear Doppler of the echo signal is separated by using the method of bimodal interference processing. Then, the processed echo signal in the TF domain is analyzed, and the phase of the processed echo signal is compensated according to the formation mechanism of the flash. Finally, the angular Doppler information is reconstructed according to the constructed compensation phase, and the number of scattering points, rotation radius, initial phase, and rotation angular velocity of the target are estimated. In addition, in order to avoid the influence of the instability of a single flash on the performance of the algorithm, this paper sequentially increases the number of observed flashes by means of iterative compensation and performs a joint search for the compensated parameters, which can effectively improve the target recognition ability of the vortex radar in specific motion scenarios. The final simulation experiments show that the method proposed in this paper has high accuracy and reliability for the separation of target angular Doppler and the estimation of characteristic parameters, which provides additional solutions for further rotor target recognition.

**Author Contributions:** Conceptualization, Y.Z. (Yongzhong Zhu) and L.B.; Formal analysis, Y.C. and L.B.; Funding acquisition, Y.Z. (Yongzhong Zhu) and Y.C.; Methodology, Y.Z. (Yongzhong Zhu) and L.B.; Project administration, X.S. and L.B.; Resources, Y.Z. (Yongzhong Zhu) and L.B.; Supervision, Y.Z. (Yongzhong Zhu); Validation, L.B.; Visualization, Y.C., L.B. and X.S.; Writing—original draft, Y.Y. and Y.Z. (Yadan Zang); Writing—review and editing, L.B., Y.Y. and Y.Z. (Yadan Zang). All authors have read and agreed to the published version of the manuscript.

**Funding:** This work was supported by the National Natural Science Foundation of China under Grant 61771490, 61801516, and the Basic Research Foundation of Engineering University of PAP under Grant WJY201908.

**Institutional Review Board Statement:** The study did not require ethical approval

**Informed Consent Statement:** The study did not involve humans.

**Data Availability Statement:** Not applicable.

**Conflicts of Interest:** The authors declare no conflict of interest.

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
