# Peer review of "Micro-Motion Parameter Extraction of Multi-Scattering-Point Target Based on Vortex Electromagnetic Wave Radar"

_remotesensing, doi:10.3390/rs14235908_

Round 1

Reviewer 1 Report

In the manuscript, authors presented a micro-motion features extraction method for spinning target with multiple scattering points based on VEMW radar, and showed the performance. For their work, several comments have been provided below:

1. For the Fig. 1, it is not very professional, such as x, y, z, which need to show using the italic; Besides, the figure is not high resolution, please improve it.

2. The number size of colorbars looks so small so that readers cannot see them clearly, please revise and increase the size of the font in colorbars, such as Fig. 3, Fig. 6(6), etc. In addition, please keep the same font size for your Figures, for example, the different font sizes are shown between Fig.8 and Fig.9, hope they could be corrected.

3. Authors presented many expressions in the manuscript, for some of them, in my view, it is not necessary to be included, however, some relevent works could be cited instead.

3. 

Reviewer 2 Report

Comments can be found in the attachment

Reviewer 3 Report

Thank you for the great work. Paper is nicely written and covers the relevant components.

Would it be possible to add a more elaborate captions to the Figures (what is shown, how it is obtained and what is a relevant feature to which the reader should pay attention). This applies especially to the multi-panel figures (e.g. Figure 4, 6, 7, 10).

Also, the color bars seem to have values, but there is no unit associated with the values. Maybe a comment on that, even if the value is unitless, it should be specified.

Round 2

Reviewer 2 Report

(1) Most of my concerns have been addressed, and the manuscript has been improved quite a lot.

(2) Although the coordinate system has been changed to right-hand system, the definition of the azimuthal angle phi_p is still questionable.
